# Is There an Advantage of Ultrathin-Strut Drug-Eluting Stents over Second- and Third-Generation Drug-Eluting Stents?

**DOI:** 10.3390/jpm13050753

**Published:** 2023-04-28

**Authors:** Flavius-Alexandru Gherasie, Chioncel Valentin, Stefan-Sebastian Busnatu

**Affiliations:** 1Department of Cardiology, University of Medicine and Pharmacy “Carol Davila,” 050474 Bucharest, Romania; stefan.busnatu@umfcd.ro; 2Emergency Clinical Hospital Dr. Bagdasar-Arseni, 050474 Bucharest, Romania

**Keywords:** eluting drug stents, ultrathin-strut eluting stents, stent thrombosis, stent restenosis, calcified coronary lesions, neointimal hyperplasia, coronary artery disease, chronic total occlusions

## Abstract

In patients undergoing percutaneous coronary intervention, the second-generation drug-eluting stents (DES) are considered the gold standard of care for revascularization. By reducing neointimal hyperplasia, drug-eluting coronary stents decrease the need for repeat revascularizations compared with conventional coronary stents without an antiproliferative drug coating. It is important to note that early-generation DESs were associated with an increased risk of very late stent thrombosis, most likely due to delayed endothelialization or a delayed hypersensitivity reaction to the polymer. Studies have shown a lower risk of very late stent thrombosis with developing second-generation DESs with biocompatible and biodegradable polymers or without polymers altogether. In addition, research has indicated that thinner struts are associated with a reduced risk of intrastent restenosis and angiographic and clinical results. A DES with ultrathin struts (strut thickness of 70 µm) is more flexible, facilitates better tracking, and is more crossable than a conventional second-generation DES. The question is whether ultrathin eluting drug stents suit all kinds of lesions. Several authors have reported that improved coverage with less thrombus protrusion reduced the risk of distal embolization in patients with ST-elevation myocardial infarction (STEMI). Others have described that an ultrathin stent might recoil due to low radial strength. This could lead to residual stenosis and repeated revascularization of the artery. In CTO patients, the ultrathin stent failed to prove non-inferiority regarding in-segment late lumen loss and showed statistically higher rates of restenosis. Ultrathin-strut DESs with biodegradable polymers have limitations when treating calcified (or ostial) lesions and CTOs. However, they also possess certain advantages regarding deliverability (tight stenosis, tortuous lesions, high angulation, etc.), ease of use in bifurcation lesions, better endothelialization and vascular healing, and reducing stent thrombosis risk. In light of this, ultrathin-strut stents present a promising alternative to existing DESs of the second and third generation. The aims of the study are to compare ultrathin eluting stents with second- and third-generation conventional stents regarding procedural performance and outcomes based on different lesion types and specific populations.

## 1. Introduction: From Coronary Balloon Angioplasty to Drug-Eluting Stent Interventions

Atherosclerotic plaques are responsible for developing coronary artery disease (CAD), which remains a leading cause of morbidity and mortality. A metallic drug-eluting stent (DES), inserted after balloon angioplasty on top of medical treatment, is frequently used to treat unstable or clinically significant coronary artery disease [1]. Andreas Gruentzig performed the first coronary angioplasty in 1977 [2]. In the early days of angioplasty, when stents were unavailable, their effectiveness was reduced by acute closure or re-stenosis. Sigwart and Puel were the first to implant a coronary stent in 1986 [3] (Palmaz-Schatz^®^; Johnson & Johnson, New Brunswick, NJ, USA). Initially, coronary stents were used to treat failures in balloon angioplasty treatment, such as acute vessel closure due to dissection or restenosis due to elastic recoil. Many other stents have become available since the beginning of 1990, including Wiktor^®^ (Medtronic, Minneapolis, MN, USA), Micro^®^ (Applied Vascular Engineering, Twickenham, UK), Cordis^®^ (Cordis, Santa Clara, CA, USA), and MULTI-LINK^®^ (Advanced Cardiovascular Systems, Santa Clara, CA, USA) [4]. There was a reduced incidence of acute vessel closure and early elastic recoil using bare metal stents (BMSs).

However, a revolutionary paradigm was born. There was a significant risk of in-stent restenosis primarily because vascular smooth muscle cells proliferated and migrated within the device [5]. Until two landmark trials changed the trajectory of coronary PCI (percutaneous coronary interventions), stents were reserved for acute or threatened closures or restenosis following balloon angioplasty. Although there is evidence to support the superiority of BMSs over balloon angioplasty [6,7] in the Belgium Netherlands Stent Arterial Revascularization Therapies Study (BENESTENT) [8] and the North American Stent Restenosis Study (STRESS) [9], it remains that 20–30% of patients experienced in-stent restenosis (ISR).

A polymer coating was used to improve the performance of coronary stents further. Compared with coronary stents without an antiproliferative drug coating, drug-eluting stents (DES) reduce neointimal hyperplasia, reducing the need for repeat revascularizations [10]. The first sirolimus-eluting stent was implanted in 1999 by Eduardo Sousa and became clinically available in 2002 as CYPHER (Cordis). Compared with BMSs, CYPHER demonstrated a significant reduction in in-stent restenosis and target vessel revascularization (TVR). Although a great step up in the evolution of PCI, it has been found that early-generation DESs are associated with an increased risk of very late stent thrombosis, probably as a result of a delayed endothelialization response to anti-restenotic drugs or a delayed hypersensitivity reaction to the polymer [11,12,13]. There has been a significant development in the design of second-generation DESs with biocompatible and biodegradable polymers, or even without polymers altogether, to reduce the risk of very late stent thrombosis. Stent thrombosis is induced by the antiproliferative effect of the DES, which delays the re-endothelialization of synthetic materials. When oral antiplatelet therapy is discontinued, the exposed scaffold surface can activate platelets, resulting in late restenosis or thrombosis [14]. Restenosis caused by neointimal hyperplasia generally occurs gradually, while thrombosis caused by stents develops abruptly and can escalate to life-threatening complications. Despite its low incidence, it is associated with a high mortality rate [15].

The development of third-generation drug-eluting stents containing biodegradable rather than durable polymers was prompted by the potential for late and very late stent thrombosis and the necessity for prolonged dual antiplatelet therapy associated with the durable polymer coating of first- and second-generation drug-eluting stents. Apart from being made from bio-degradable polymers, most of these new stents are also made from cobalt–chromium or platinum–chromium platforms, especially in ultrathin struts, and a few have abluminal polymers.

The trials that compared second-generation DESs were primarily designed as non-inferiority studies, and the most significant tests used the Xience stent as a comparator [16].

There have been concerns about hypersensitivity to stents, and biodegradable or polymer-free materials have been developed for stents. Drug-eluting stents differ in their characteristics. They differ in their drug-loading abilities, drug-release pharmacokinetics, polymer strength, chemical compatibility, and impact on vascular wall thinning, aneurysm development, and late restenosis. Following stent implantation, various complications may occur. Biocompatibility questions arise when components are not native to the human body. Corrosion issues and toxic substances emitted during corrosion have led to the development of biocompatible and biodegradable materials. In the early stages of biocompatibility, problems such as thrombosis, inflammation, and the development of neointima may occur.

Scaffold fracture is a progressive condition linked with poor biocompatibility [17]. One of the main risk factors for stent fractures is the length of the stent (5 cm versus 3 cm) and the positioning of the stent in a bypass graft or the right coronary artery [18,19]. The conservative treatment of stent fractures without restenosis has shown favorable results [19]. There is a problem known as malposition, which occurs when the struts of the stent are not aligned with the vessel surface. The condition occurs in two to five percent of cases and is a leading cause of late stent thrombosis [18].

The metallic backbone of the DES is a potential target for further development. It plays a critical role in handling the devices and their safety and effectiveness. It has been demonstrated that patients are more likely to have improved clinical outcomes after percutaneous coronary intervention (PCI) if stent strut thickness is reduced, such as in ultrathin stent struts (≤70 μm).

Regarding the potential cost implications of ultrathin drug-eluting stents compared with conventional DESs, we can summarize that a second-generation DES’s average cost is 150 dollars, and drug-eluting stents cost almost double.

The aims of the study are to compare ultrathin eluting stents with second- and third-generation conventional stents regarding procedural performance and outcomes based on different lesion types and specific populations.

## 2. A View of Drug-Eluting Stent Components

A modern stent contains four primary characteristics: its metallic platform, its strut thickness, its polymer coating thickness, and its polymer type [20]. The physical features of the stent platform play an essential role in deliverability and restenosis limitation. Various technical stent specifications, such as the type of expansion, material, surface smoothness, strut thickness, and shape, have been identified as the primary triggers contributing to restenosis. Cypher and Taxus are two of the first-generation DESs, which are manufactured from 316L stainless steel with balloon-expandable systems. Stents with thicker struts are more radiopaque and have sufficient radial strength. However, they are more likely to develop restenosis than those with thinner struts [21]. Furthermore, due to its low density and ferromagnetic nature, 316L stainless steel is incompatible with MRI and is not easily visible under fluorescence.

Xience V (everolimus-eluting stent, Abbott Vascular, Santa Clara, CA, USA) and Endeavor (zotarolimus-eluting stent, Medtronic Vascular, Santa Rosa, CA, USA) were the second generation of DESs that employed cobalt chromium (CoCr) with thinner struts. In addition to reducing neointimal responses and increasing the rate of endothelialization, these results were achieved [22].

### 2.1. Metallic Platform Designs

A stent can be categorized based on its metallic basis and design. Stent scaffolds can be made from various metallic compounds, which are further distinguished by their distinct physical attributes, including radial strength, ferromagnetism, durability, and radiolucency. Radial strength refers to external force resistance. Radiolucency describes the stent’s appearance when undergoing angiography [23]. In the beginning phases of BMS, stainless steel or tantalum was used. The angiography of tantalum is straightforward and does not exhibit any ferromagnetism. Tantalum undergoes oxidation after implantation, increasing stability and degradation resistance [24]. The thrombogenicity of tantalum was similar to that of stainless steel, but its radio opacity was superior to stainless steel [25]. Tantalum is appreciated for its mechanical strength. In contrast, 316L stainless steel combines iron, chromium, and nickel. The backbones of recent stents are composed of chromium, cobalt, or platinum alloys to allow a thin structure for the struts. A cobalt chromium stent is an excellent solution for severe lesions due to its high radial strength, slim profile, and improved elasticity [26]. Platinum–chromium stents are widely recognized for their superb flexibility, delivery capability, adaptability, radial force, and ease of visualization [27].

Two trials have shown that thinner strut designs will improve long-term performance. According to ISAR-STEREO, the incidence of angiographic restenosis in the narrow strut group was 15.0%, whereas the incidence of angiographic restenosis in the thick-strut group was 25.8% (relative risk, 0.58; 95% confidence interval, 0.39 to 0.87; *p* = 0.003). There was no difference between the death and myocardial infarction rates over the first year [28]. In the ACS RX Multi-Link trial, the thin-strut group showed a mortality rate of 1.5% compared with 2.5% for the thick-strut group [29]. Thinner struts demonstrate better clinical results because they may decrease inflammation and vascular injury, accelerate endothelialization, decrease neointimal proliferation, and decrease thrombogenicity since they provide less contact surface for body cells to produce these responses [30]. For confirmation, the ISAR-STEREO-2 trial results were presented in 2003, which enhanced the ISAR-STEREO results [31]. It was demonstrated in both trials that the thinner struts improved stent quality and reduced complications. A newly developed metallic platform for stents, cobalt–chromium (CoCr) and platinum–chromium (PtCr), was designed to keep radial strength and recoil characteristics intact. Table 1 reviews the specifications of stent components.

Compared with stainless steel, cobalt–chromium is a denser metal with enhanced qualities best suited to stent development. In contrast, platinum–chromium is a unique base metal developed for improving stent quality. Compared with stainless steel, CoCr and PtCr reduce the thickness of struts by up to 70 µm.

### 2.2. Drugs

In the second-generation stent, new Limus drugs were incorporated. Sirolimus derivatives, Zotarolimus and Everolimus, have the same structure as Sirolimus, but their pharmacological properties vary. The semi-synthetic drug, Zotarolimus, has a similar mechanism of action to Sirolimus. However, its chemical composition differs due to the insertion of a tetrazole ring at position 42 of the native molecule [33].

As a result of this modification, Zotarolimus becomes the most lipophilic and water-repellent of all Sirolimus analogs. Rather than a burst release, this modification allows Zotarolimus to release over a sustained period [33,34].

In second-generation stents, everolimus-eluting stents (EES) are widely recognized as the most effective. Everolimus is also a semi-synthesized hydroxyethyl ether derivative of Sirolimus, which has been modified by replacing the hydroxyl group at position 40 with a 2-hydroxyethyl group [35]. Comparatively, Everolimus is polar and more lipophilic than Sirolimus, facilitating improved bioavailability, a shorter clearance time, a longer cellular residence time, and sound absorption [35]. The third-generation DES contains all members of the Limus family. The Limus family was further expanded with the addition of BioLimus. Terumo and Biosensors International introduced Biolimus, a macrocyclic lactone derivative of Sirolimus. This has similar effects to Sirolimus, in terms of anti-inflammation and anti-proliferation, but the release kinetics of this drug are superior to those of Sirolimus [36,37].

### 2.3. Polymers

For drug-eluting stents, the formula and nature of the polymer have become crucial elements since the drug-release kinetics are controlled by it. It has been noted that the first generation of drug-eluting stents did achieve excellent results; however, some issues, such as the uncontrolled proliferation of neointimal cells and inflammation, were encountered, and the nature of the polymer was primarily responsible for these responses. The second generation of stents used new and different polymers to achieve better results. Medtronic developed a Phosphoryocholine polymer consisting of four separate monomers of Methacrylate (2-methacryloyloxyethyl, lauryl methacrylate, trimethoxysilylpropyl methacrylate, and 2-hydroxypropyl-methacrylate. It is naturally present in our membrane, is highly hydrophilic, and does not stimulate inflammatory responses [38]. Abbott Laboratories launched a polymer composed of fluorinated copolymer poly(vinylidene fluoride)-co-hexafluoropropylene (PVDF-HFP), which could tolerate expansion by maintaining its elastic flexibility [39].

Coronary percutaneous interventions currently treat lesions that were not considered to have an interventional solution in the past; with our debulking solution, an interventional approach could treat most calcified lesions. Because intravascular lithotripsy (IVL) is used for the preparation of the vessel before stenting and for stent under expansion in calcified lesions, there is a concern regarding the effect of using IVL on the polymer coating. Achim et al. showed by electron microscopic analysis that IVL interaction with DES’s fluoropolymer does not significantly reduce its antiproliferative properties. It also showed that forcing the stent into a narrow, calcified stenosis can result in identical polymer degradation [40].

Biocompatible and biodegradable polymers are used to coat third-generation stents. Currently, polylactic acid (PLA) is the most commonly used polymer, followed by poly-d,l-lactic acid (PDLLA), polyglycolide (PLG), and poly(D,L-lactic-co-glycolic acid) (PLGA). As these polymers degrade under hydrolytic conditions and are biocompatible, they cause little harm when their released constituents are released. Polylactic acid is a monomer of L-lactide or D-lactide. PLA degradation is determined by several elements, specifically the surface area to volume ratio, coating thickness, and porosity. A biodegradable polymer breaks down into carbon, methane, water, and biomass after decomposition [41]. To prevent coating deficiency in the targeted area, Terumo introduced a gradient coating in which the areas that experienced the maximum physical stress were left bare of coating to mitigate potential inflammatory reactions in the long run [42]. Struts were only coated in the middle to avoid cracking. Boston Scientific developed a novel solution by introducing the thin coating strategy. It covered the stents abluminal with a gradient reduction in coating thickness from the core section down to the sides [43].

## 3. The Advantages of Ultrathin-Strut Stents in Early Vessel Healing

Due to the antiproliferative drugs used in first-generation DESs, fewer patients had neointimal hyperplasia, but vascular healing was impaired, leading to late and incomplete endothelialization. On top of that, there may be a delay in recovery due to hypersensitivity reactions caused by permanent polymers. As a result, the blood flow is exposed to thrombogenic struts, which can lead to stent thrombosis [44].

In a rabbit denudation model presented by Soucy et al., strut coverage at day 14 was as high as 95% in the thinnest struts (81 μm) and lower with thicker struts: 88% in stents with 97 μm struts and 77% with 132 μm struts [45]. Based on an in vivo optical coherence tomography (OCT) study in a porcine model, the thinnest strut stent (61 μm) achieved faster and better strut coverage [39]. Endothelialization may be delayed, and the risk of restenosis increases with thicker struts due to the larger surface taking longer to endothelial. Moreover, thicker struts may contribute to more significant vessel injury and inflammation in adjacent tissue due to the penetrating struts’ traumatic disruption of the internal elastic lamina. As a result of increased intimal inflammation, neo-intimal growth and hyperplasia lead to restenosis.

The ISAR-STEREO trial demonstrated that strut thickness affects ISR rates [28]. There were 651 patients randomized to receive either a thin-strut stent (50 μm) or a comparable stent with a strut thickness of 140 μm, without a polymer or antiproliferative agent. There was almost twice as much angiographic restenosis (defined as >50% stenosis at 6-month follow-up angiography) in the thick-strut group compared with the thin-strut group (15.0% versus 25.8%, respectively).

Rittersma et al. also suggested that thinner struts are associated with a decreased risk of ISR and a reduced risk of angiographic and clinical restenosis [46].

## 4. The Advantages of Ultrathin-Strut Stents in Deliverability of Drug-Eluting Stents

Data on mechanical behavior during delivery of the stent is limited, even though there are numerous studies on deployment and especially in vivo function. Many different delivery systems are available for stent placement, and the deliverability is determined by pushability, trackability, and crossability [47].

Compared with conventional second-generation DESs, ultrathin struts may be more flexible, improve trackability, and have a lower profile, improving crossability [48].

## 5. A Negative Influence of Ultrathin Struts on the Mechanical Properties of Drug-Eluting Stents

Coronary stents should have an excellent radial force to maintain lumen patency to resist high external pressures. Generally, this pressure is around 200 mmHg in a healthy coronary artery and much more significant in a calcified lesion. It is possible to develop ISR due to failure to resist (chronic) external pressure [49]. Bonin et al. suggested that a stent’s resistance to external forces is determined by its radial stiffness, which occurs when uniform external radial forces are applied, and its radial strength, which is determined by the pressure that permanently deforms the stent [47]. As a result of its higher elastic modulus and tensile strength, cobalt–chromium has a higher radial strength than stainless steel, thereby allowing thinner struts to be used without sacrificing radial strength.

The majority of contemporary stents are modular, consisting of undulated rings. These rings are connected by connectors, which provide longitudinal support to the stent. In coronary stent design, the number and orientation of connectors between rings play a critical role in determining mechanical properties. Compared with an open-cell structure, closed-cell designs (more connectors) provide better vessel wall coverage and are likely to prevent plaque prolapse [50]. Several authors reported that improved coverage with less thrombus protrusion reduced the risk of distal embolization in patients with ST-elevation myocardial infarction (STEMI) [51].

Open-cell designs, however, are more flexible, deliverable, and conformable than closed-cell designs. A further advantage of these devices is that they facilitate more accessible access to side branches in cases of bifurcation lesions.

There is a possibility that a stent may recoil due to low radial strength. Acute stent recoil leads to residual stenosis and repeated artery revascularization [52,53,54]. According to an observational study conducted on 128 patients who underwent PCI for chronic total occlusions (CTOs), the ultrathin-strut Orsiro stent was associated with higher absolute (measured in millimeters) and relative (measured in percent) recoil than the Resolute Onyx zotarolimus-eluting stent with 81 μm struts (Medtronic Cardiovascular, Santa Rosa, CA, USA) [53].

Teeuwen et al. demonstrated in the randomized PRISON IV trial, which compared the Orsiro stent with the Xience stent in CTO patients, that the Orsiro ultrathin stent failed to prove non-inferiority in terms of in-segment late lumen loss and did show statistically higher rates of restenosis [55].

Overall, radial strength is determined by the type of metal used, the strut’s thickness, and the stent’s design. While cobalt–chromium and platinum–chromium alloy struts have comparable radial strength to stainless steel struts, some clinical data indicate that the radial strength of ultrathin-strut stents may not be sufficient to treat CTO lesions and may result in a more significant stent recoil than conventional second-generation DESs. Different manufacturers are developing ultrathin-strut stents. Table 2 reviews the most commonly used features [48].

## 6. Patient-Specific Stents

In most stent trials, there is no evidence of an interaction between patient subgroups and the efficacy of one type of stent compared with another. A biodegradable polymer DES may be more suitable for patients with stent thrombosis myocardial infarction [56]. The BIOSTEMI trial demonstrated the superiority of the ultrathin sirolimus-eluting bioresorbable polymer stent Orsiro over the durable polymer everolimus-eluting XIENCE at one year based on TLF [57].

A post hoc subgroup comparison study from the BIOSTEMI randomized trial showed that ultrathin-strut biodegradable polymer sirolimus-eluting stents performed better than thin-strut durable polymer everolimus-eluting stents regarding the target lesion failure at two years among STEMI patients having undergone both complex and noncomplex primary percutaneous coronary intervention [58].

Based on the initial results from multicenter trials, the Supraflex Cruz stent has shown promising clinical outcomes in a wide range of lesions and clinical parameters. The talent trial and Flex registry showed a low incidence of TLR and stent thrombosis [59,60]. ORIENT is a randomized controlled trial that compared the angiographic results of stents Orsiro and Resolute Integrity TM in 372 patients across eight South Korean centers [61]. At the 9-month follow-up, the primary endpoint was non-inferiority for in-stent late lumen loss. Orsiro was non-inferior to Resolute Integrity (median late lumen loss was 0.06 mm and 0.12 mm, respectively; *p* for superiority = 0.205). There is a similar clinical outcome between the Orsiro and Resolute Integrity groups regarding the TLF endpoint (composite of cardiac death, target-vessel MI, and TLR); however, the study was insufficiently powered to assess clinical outcomes.

In a three-year clinical follow-up assessment, the TLF rates were 4.7% (Orsiro) and 7.8% (Resolute Integrity) (*p* = 0.232). A total of two patients were diagnosed with stent thrombosis in the Resolute integrity group compared with none in the Orsiro group (0.0% versus 1.6%, respectively; *p* = 0.040) [62].

Among patients with high bleeding risk, BioFreedom and Resolute Onyx with 1-month dual antiplatelet therapy (DAPT) seem to provide the best results [63]. Certain stents may perform better or have been extensively studied to treat some complex lesions, such as left main stem stents, bifurcation stents, chronic total occlusions, long lesions, and calcified lesions. Several dedicated trials have been conducted with everolimus-eluting stents to treat left central disease or chronic total occlusions without protection [64,65,66].

It was reported that the BIOFLOW-V trial, which involved 1334 patients, found that target lesion failure, cardiac death, and clinically driven target lesion revascularization were lower, but not statistically different, in patients treated with ultrathin-strut bioresorbable polymer siro-limus-eluting stents (BP SES) as compared with thin-strut durable polymer everolimus-eluting stents (DP EES) at five years. The risk of target vessel myocardial infarction (TV-MI) and late definite/probable stent thrombosis was substantially reduced for patients treated with BP SES [67].

A study of the efficacy of Orsiro in treating CTO lesions was completed in Prison IV, where 330 subjects were randomized 1:1 to the Orsiro SES or the Xience EES. Even though the trial was underpowered for clinical performance, it failed to demonstrate the non-inferiority of the Orsiro SES for LLL at nine months. After three years of follow-up, the Orsiro SES arm showed a higher MACE rate than the comparative arm. However, the subgroup analysis of patients with CTO who participated in the BIOFLOW III and SORT-OUT VII trials revealed a low TLF and TLR [68].

DESs with good radial strength and visibility would be required for ostial lesions. In cases of calcified or tortuous lesions, reasonable radial force and deliverability will be beneficial. Delivery is easier with DESs consisting of thinner struts and fewer connectors between stent rings.

Research is being undertaken to optimize the design of DESs, and the efficacy of these systems in real-world scenarios as the search for the ideal stent continues. However, there is limited evidence regarding the safety and effectiveness of ultrathin-stent DESs in left main or coronary bifurcations. Most revascularization procedures will be conducted with ultrathin stents after safety data are available in real-life patient populations. Shortly, research in stent design will likely focus on developing more biocompatible drugs, alloys, and polymers. Future DESs are expected to be highlighted by a gradual progress in deliverability and flexibility.

## 7. Conclusions

Clinical trials with DESs with ultrathin struts have demonstrated promising outcomes for further improving clinical outcomes after percutaneous coronary angioplasty interventions. The Orsiro stent is superior to other non-ultrathin second-generation DESs in patients with STEMI. However, ultrathin-strut stents may perform less favorably in some lesions, such as those that are heavily calcified or CTOs, than their second-generation counterparts with thicker struts. Further studies will be needed to clarify this.

Therefore, ultrathin-strut DESs with biodegradable polymers have some limitations in calcified (or ostial) lesions and CTOs; however, they have certain advantages in terms of better deliverability (tight stenosis, tortuous lesions, high angulation, etc.), easy use in bifurcation lesions, better endothelialization and vascular healing, and reduction in the risk of stent thrombosis.

## Figures and Tables

**Table 1 jpm-13-00753-t001:** Specifications of stent components; table adapted from [32].

Properties	Elastic Modulus (GPa)	Tensile Strength (MPa)	Yield Strength (MPa)	Density (g/cm^3^)
Stainless steel 316L	190	535	275	7.9
PtCr	203	834	480	9.9
CoCr (mp35)	233	930	414	9.1
CoCr (L605)	243	1000	500	8.4

**Table 2 jpm-13-00753-t002:** A brief review of mainly used ultrathin-strut stents.

Stent Name	Orsiro	Biomime	CoroFlex IsarNeo	Supraflex	Evermine 50	MiStent
Company	Biotronik	Merril	B.Braun	SMT	Meril	Micell
Material	Cobalt-chromium	Cobalt-chromium	Cobalt-chromium	Cobalt-chromium	Cobalt-chromium	Cobalt-chromium
Strut thickness (μm)	60	65	55	60	50	64
Coating distribution	Circumferential	Circumferential	**-**	Circumferential	Circumferential	Circumferential
Polymer	Biodegradable	Biodegradable	Polymer free	Biodegradable	Biodegradable	Biodegradable
Eluting drug	Sirolimus	Sirolimus	Sirolimus	Sirolimus	Everolimus	Sirolimus
Drug dose	1.4 μg/mm^2^	1.25 μg/mm^2^	1.2 μg/mm^2^	1.4 μg/mm^2^	1.25 μg/mm^2^	2.4 μg/mm^2^
Drug release	50% by 1 month 80% by 3 months	30–40 days	80% by 28 days 100% by 90 days	70% by 1 week 100% by 3–4 months	30–40 days	9 months

## Data Availability

Not applicable.

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
