# Peer review of "Is There an Advantage of Ultrathin-Strut Drug-Eluting Stents over Second- and Third-Generation Drug-Eluting Stents?"

_jpm, 2023, doi:10.3390/jpm13050753_

Round 1

Reviewer 1 Report

The novelty and relevance of the review are questionable. Most of the references are 10-20 years old. The authors go too much into details of very old studies of no relevance any more and discuss a large number of studies superficially. The distinction between second-third generation DES and ultrathin-strut DES is not clear. 

The quality of English of the paper is appropriate. Only few typos were found.

Author Response

Thank you for your answer. I revised the references and recent trials and also made clearer the differences between the second-third generation DES and ultrathin-strut DES.

The development of third-generation drug-eluting stents containing biodegradable rather than durable polymers was prompted by the potential for late and very late stent thrombosis and the necessity for prolonged dual antiplatelet therapy associated with the durable polymer coating of first and second-generation drug-eluting stents. Apart from being made from bio-degradable polymers, most of these new stents are also made from cobalt-chromium or plate-num-chromium platforms, especially in ultra-thin struts, and a few have abluminal polymers.

The trials that compared second-generation DES were primarily designed as non-inferiority studies, and the most significant trials used the Xience stent as a comparator.

Reviewer 2 Report

The article provides a comprehensive overview of the advantages and limitations of ultrathin drug-eluting stents (DES) in percutaneous coronary intervention (PCI) and compares them with conventional DESs. The use of second-generation DESs with biocompatible and biodegradable polymers has reduced the risk of very late stent thrombosis. However, there are limitations to using ultrathin-strut DESs in treating calcified or ostial lesions and chronic total occlusions (CTOs), and their effectiveness in these situations is mixed. The authors could have included the potential cost implications of using ultrathin drug-eluting stents compared to conventional DESs.

Author Response

Thank you for your expertise. Regarding the potential cost implications of ultrathin drug-eluting stents compared to conventional DESs, we can summarize that a second-generation DES's average cost is 150 dollars, and drug-eluting stents cost almost double. I added this information to the paper.

Reviewer 3 Report

This interesting yet important work highlights the importance of current ultra-thin strut stents as a promising alternative to existing DES. I recommend its acceptance with minor modifications, such as more evidence from previous data about its superior usage.

Author Response

Thank you for your expertise; I added more recent studies.

Reviewer 4 Report

In this paper, the author reviewed the advantages of second-generation drug-eluting stents (DES) in coronary artery intervention treatments. These DESs have biocompatible and biodegradable polymers, reducing the risk of late stent thrombosis. However, there is uncertainty about whether ultra-thin drug-eluting stents are suitable for all lesion types. Despite limitations in treating calcified lesions and CTOs, ultra-thin strut DESs with biodegradable polymers have advantages in deliverability, ease of use in bifurcation lesions, better endothelialization, and vascular healing, reducing stent thrombosis risk. Therefore, ultra-thin strut stents hold great potential as an alternative to existing second and third-generation DESs. However, some issues may limit the strength of the conclusions, which need to address.

Comments:

1.     There are some grammar mistakes; please correct.

The English Language needs to be modified, and some grammar mistakes need to be corrected. 

Author Response

I appreciate your comments, the paper is having a complete English correction. Thank you!